# Super Formula for Diagnosing Disseminated Intravascular Coagulation Using Soluble C-Type Lectin-like Receptor 2

**DOI:** 10.3390/diagnostics13132299

**Published:** 2023-07-06

**Authors:** Akitaka Yamamoto, Hideo Wada, Masaki Tomida, Yuhuko Ichikawa, Minoru Ezaki, Katsuya Shiraki, Motomu Shimaoka, Toshiaki Iba, Katsue Suzuki-Inoue, Masahide Kawamura, Hideto Shimpo

**Affiliations:** 1Department of Emergency and Critical Care Center, Mie Prefectural General Medical Center, Yokkaichi 510-8561, Japan; akitaka-yamamoto@mie-gmc.jp (A.Y.); st25053@yahoo.co.jp (M.T.); 2Department of General and Laboratory Medicine, Mie Prefectural General Medical Center, Yokkaichi 510-8561, Japan; katsuya-shiraki@mie-gmc.jp; 3Department of Central Laboratory, Mie Prefectural General Medical Center, Yokkaichi 510-8561, Japan; ichi911239@yahoo.co.jp (Y.I.); ajbyd06188@yahoo.co.jp (M.E.); 4Department of Molecular Pathobiology and Cell Adhesion Biology, Mie University Graduate School of Medicine, Tsu 514-8507, Japan; motomushimaoka@gmail.com; 5Department of Emergency and Disaster Medicine, Juntendo University Graduate School of Medicine, Tokyo 113-8431, Japan; toshiiba@juntendo.ac.jp; 6Department of Clinical and Laboratory Medicine, Faculty of Medicine, University of Yamanashi, Yamanashi 409-3898, Japan; katsuei@yamanashi.ac.jp; 7Department of Research and Development, IVD Business Segment, LSI Medience Corporation, Tokyo 174-8555, Japan; kawamura.masahide@mv.medience.co.jp; 8Mie Prefectural General Medical Center, Yokkaichi 510-8561, Japan; hideto-shimpo@mie-gmc.jp

**Keywords:** DIC, pre-DIC, sCLEC-2, sCLEC-2xD-dimer/PLT

## Abstract

The scoring systems for disseminated intravascular coagulation (DIC) criteria require several adequate cutoff values, vary, and are complicated. Accordingly, a simpler and quicker diagnostic method for DIC is needed. Under such circumstances, soluble C-type lectin-like receptor 2 (sCLEC-2) received attention as a biomarker for platelet activation. Materials and Methods: The diagnostic usefulness of sCLEC-2 and several formulas, including sCLEC-2xD-dimer, sCLEC-2/platelet count (sCLEC-2/PLT), and sCLEC-2/PLT × D-dimer (sCLEC-2xD-dimer/PLT), were evaluated among 38 patients with DIC, 39 patients with pre-DIC and 222 patients without DIC or pre-DIC (non-DIC). Results: Although the plasma level of sCLEC-2 alone was not a strong biomarker for the diagnosis of DIC or pre-DIC, the sCLEC-2xD-dimer/PLT values in patients with DIC were significantly higher than those in patients without DIC, and in a receiver operating characteristic (ROC) analysis for the diagnosis of DIC, sCLEC-2xD-dimer/PLT showed the highest AUC, sensitivity, and odds ratio. This formula is useful for the diagnosis of both pre-DIC and DIC. sCLEC-2xD-dimer/PLT values were significantly higher in non-survivors than in survivors. Conclusion: The sCLEC-2xD-dimer/PLT formula is simple, easy, and highly useful for the diagnosis of DIC and pre-DIC without the use of a scoring system.

## 1. Introduction

Disseminated intravascular coagulation (DIC) is a fatal disease that is often complicated by major bleeding or organ failure [1,2,3], although there are many guidelines for the management of DIC [4,5,6]. The main underlying diseases are infectious diseases, hematological malignancy, solid cancer, obstetrics, trauma, and aortic aneurysm [1,2,3]. DIC patients are generally treated for underlying diseases, and early treatment with anticoagulants such as antithrombin [1,4,7,8] or recombinant thrombomodulin [9,10,11] has been recommended for DIC in Japan.

As there is no gold standard for the diagnosis of DIC, many diagnostic criteria for DIC have been established by the Japanese Ministry of Health, Labour and Welfare (JMHLW) [12], the International Society of Thrombosis Haemostasis (ISTH) [13], the Japanese Association for Acute Medicine (JAAM) DIC [14], and the Japanese Society of Thrombosis Hemostasis [15]. Most diagnostic criteria for DIC are based on scoring systems for global coagulation tests, such as fibrin-related markers (FRMs), prothrombin time (PT), platelet count, and fibrinogen [15,16,17,18]. FRMs, including fibrinogen and fibrin degradation products (FDPs), D-dimer or soluble fibrin (SF), and D-dimer, require standardization [19,20,21]. The use of scoring systems for DIC is complicated, and multiple cutoff values for the parameters of DIC vary among the four diagnostic criteria for DIC [12,13,14,15]. Four DIC groups that were diagnosed using each of the diagnostic criteria showed variation in their disease severity and outcomes [22]. 

Simpler diagnostic criteria are required to facilitate the early treatment of DIC in the emergency room (ER) or intensive care unit (ICU). Therefore, the sepsis-induced coagulopathy (SIC) score [23], which includes the platelet count and sequential organ failure assessment score [24] or quick DIC score, which includes the D-dimer levels, platelet count, PT ratio, and underlying disease [25], which were recently developed, may be simpler and easier scoring systems and may be useful for diagnosing DIC or coagulopathy in patients with sepsis [26]. However, these scoring systems need to use adequate cutoff values and cannot be compared to other diagnostic criteria. Accordingly, these scoring systems are still complicated. In addition, there is no useful biomarker for platelet activation, which plays an important role in the progression of various pathogenic states [27,28]. Recently, elevated soluble C-type lectin-like receptor 2 (sCLEC-2) levels have been reported in patients with DIC [29], thrombotic microangiopathy (TMA) [30], acute myocardial infarction [31,32], acute cerebral infarction [33], and coronavirus disease 2019 [34].

In this study, we developed a simple formula for the diagnosis of DIC using sCLEC-2, platelet count, and D-dimer to examine the agreement with the JMHLW diagnostic criteria.

## 2. Materials and Methods

The study population included patients with the following conditions who were managed at Mie Prefectural General Medical Center from 1 September 2019 to 28 December 2022: infectious diseases (*n* = 215), solid cancer (*n* = 27), aortic aneurysm (*n* = 37), hematological disorders (*n* = 11), trauma (*n* = 41), cardiopulmonary arrest (*n* = 25), and unidentified clinical syndrome (*n* = 75) (Table 1). DIC was diagnosed using the Japanese Ministry of Health Labour and Welfare criteria for DIC [12]. Patients with a DIC score of ≥7 points, 5 or 6 points, and ≤4 points were diagnosed with DIC, pre-DIC, and non-DIC, respectively.

Plasma was prepared by two centrifugations at 3000 rpm for 15 min (the platelet count was less than 0.5 × 10^10^ platelet count/L). Plasma sCLEC-2 levels were measured by a chemiluminescent enzyme immunoassay (CLEIA) using previously described monoclonal antibodies and the STACIA CLEIA system (LSI Medience, Tokyo, Japan) [31,33]. FDP, D-dimer, and SF were measured using LPIA FDP-P, LPIA-Genesis, and Iatro SF II (LSI Medience, Tokyo, Japan), respectively, with the STACIA system (LSI Medience). The activated partial thromboplastin time (APTT) and PT-international normalized ratio (INR) were measured by a Thrombocheck APTT-SLA and Thromborel S (Sysmex Co., Kobe, Japan, respectively) using an automatic coagulation analyzer CS-5100 (Sysmex Co.). The platelet counts were measured using a fully automatic blood cell counter XN-3000 (Sysmex Co.). “sCLEC-2xD-dimer”, “sCLEC-2/platelet number (sCLEC-2/PLT)” and “sCLEC-2xD-dimer/platelet number (sCLEC-2xD-dimer/PLT)” were calculated using sCLEC-2 (ng/L), platelet number (×10^10^/L) and D-dimer (μg/mL). 

### Statistical Analyses

The data are expressed as median (25th–75th percentiles). The significance of differences between groups was examined using the Mann–Whitney U-test. The cutoff values, areas under the curve (AUCs), sensitivity, specificity, and odds ratios were determined by a receiver operating characteristic (ROC) analysis; *p*-values < 0.05 were considered to indicate a statistically significant difference. All statistical analyses were performed using the Stat-Flex software program (version 7; Artec Co., Ltd., Osaka, Japan). 

## 3. Results

The mortality rate was highest in CPA and was > 10.0% in cases with solid cancer and infection and 0% in cases with hematological malignancy and UCS (Table 1). The APTT was significantly longer in cases with infection and CPA than in UCS, and the PT was significantly higher in cases with solid cancer, aortic aneurysm, infection, and CPA than UCS. Platelet counts were significantly lower in cases with aortic aneurysm, infection, and CPA than in those with UCS. In all underlying diseases, the DIC score, FDP, D-dimer, SF, and sCLEC-2 were significantly higher than in UCS. 

Regarding evaluation using the JMHLW diagnostic criteria, 38, 39, and 222 patients were diagnosed with DIC, pre-DIC, and non-DIC, respectively. (Table 2). FDP, D-dimer, SF PT-INR, and sCLEC-2 levels were significantly higher in DIC and pre-DIC than in non-DIC, platelet counts were significantly lower in DIC and pre-DIC than in non-DIC, and APTT was significantly longer in DIC and pre-DIC than in non-DIC.

In the ROC analysis (Table 3), the cutoff values for FDP and D-dimer showed the highest AUC and sensitivity (both DIC vs. non-DIC and pre-DIC vs. non-DIC). With regard to the PT-INR, APTT, platelet count, and SF, the AUC for DIC was ≥0.89 but the AUC for pre-DIC was ≤0.75. Using sCLEC-2, the AUCs for DIC (0.801) and pre-DIC (0.748) were not significantly different. 

Although the difference of sCLEC-2 alone between DIC (median value, 423 ng/L) and non-DIC (median value, 237 ng/L) and between pre-DIC (median value 361 ng/L) and non-DIC were significant, the difference of sCLEC-2 alone between DIC and pre-DIC was not significant. The differences in sCLEC-2xD-dimer/PLT (median value of DIC, Pre-DIC, and non-DIC; 128, 35.6, and 3.45, respectively), sCLEC-2/PLT (median value; 4.78, 2.37, and 1.12, respectively), or sCLEC-2xD-dimer (median value; 13,198, 6923, and 657, respectively) among DIC, pre-DIC, and non-DIC were significant. (Table 4 and Figure 1). 

In the ROC for both DIC vs. non-DIC (Figure 2) and DIC + pre-DIC vs. non-DIC (Figure 3), sCLEC-2xD-dimer/PLT showed the highest curve, and sCLEC-2xD-dimer and sCLEC-2/PLT showed high curves, while sCLEC-2 alone showed the lowest curve.

Regarding the ROC analysis (Table 5), sCLEC-2xD-dimer/PLT showed the highest AUC, sensitivity (specificity), and odds ratio for DIC (0.993, 94.7%, and 315.0, respectively), DIC + pre-DIC (0.961, 89.6%, and 74.6), and pre-DIC (0.929, 85.6%, and 32.7, respectively). sCLEC-2xD-dimer and sCLEC-2/PLT also showed high AUC, sensitivity (specificity), and odds ratio for DIC, DIC + pre-DIC, and pre-DIC.

The APTT was significantly longer in non-survivors than in survivors, and PT-INR, FDP, D-dimer, SF, and sCLEC-2 were significantly higher in non-survivors than in survivors, while platelet counts were significantly higher in non-survivors than in survivors (Table 6). sCLEC-2xD-dimer/PLT (median value, 46.3 vs. 4.29), sCLEC-2/PLT (median value, 2.53 vs. 1.25), sCLEC-2xD-dimer (4962 vs. 771), and sCLEC-2 (365 ng/L vs. 247 ng/L) were significantly higher in non-survivors than in survivors (Figure 4).

According to the ROC curve, the usefulness of the formulas for predicting a poor outcome was ranked in the order of sCLEC-2xD-dimer/PLT, sCLEC-2xD-dimer, sCLEC-2/PLT, and sCLEC-2 (Figure 5). The ROC analysis revealed that FDP, sCLEC-2xD-dimer/PLT, and PT-INR had high AUC (0.840, 0.816, and 0.817) and sensitivity (76.8%, 76.8%, and 70.6%) for predicting a poor outcome (Table 7).

## 4. Discussion

sCLEC-2 has been introduced as a new biomarker of platelet activation [35]. Elevated sCLEC-2 levels have been reported without thrombocytopenia in patients with acute coronary syndrome [31,32] or acute cerebral infarction [33], suggesting that sCLEC2 may reflect platelet activation in atherosclerotic thrombosis. In addition to reports related to platelet activation [34] such as DIC [29], TMA [30], and hypercoagulability such as nephrotic syndrome [36] and colorectal cancer [37], it was reported that CLEC-2 regulates inflammatory reactions [38,39,40] and CLEC-2 may be related to cancer progression with platelet activation and hypercoagulability [37,41]. In particular, the plasma sCLEC-2 levels in patients with COVID-19 infections were significantly higher than those in patients with other infections and reflected the progression of the severity of COVID-19 infections [34]. In particular, the sCLEC-2/platelet ratio is useful for evaluating the severity of COVID-19 infections. Furthermore, the plasma sCLEC-2 levels in patients with mild-stage COVID-19 infections were similar to those in patients with severe other pneumonia [34,42]. 

As there is no gold standard and many diagnostic criteria are used for DIC, this study evaluated hemostatic biomarkers for the diagnosis of DIC based on the diagnostic criteria of the JMHLW [12]. This is because JMHLW is the most frequently used and famous diagnostic criterion for DIC in Japan. The mortality rate of DIC diagnosed using the JMHLW criteria was higher in comparison to DIC diagnosed using the JAAM criteria [14] or SIC criteria [23], suggesting that the severity of pre-DIC in patients diagnosed according to the JMHLW criteria may be similar to the severity of that in patients diagnosed according to the JAAM or SIC criteria [22]. In addition, most of the previous diagnostic criteria for DIC have involved complicated scoring systems that require adequate cutoff values of biomarkers and scoring systems [12,13,14,15]. Biomarkers, especially D-dimer, strongly require standardization [19]. 

sCLEC-2 alone was not more useful than FDP, D-dimer, or PT-INR in the diagnosis of DIC or pre-DIC. In this study, the most useful diagnostic markers for DIC were FDP and D-dimer. These findings may depend on the underlying diseases, as this study included many patients with infectious diseases with non-DIC who had low FDP or D-dimer levels [22]. Hematological malignancy patients without DIC had relatively high FDP or D-dimer levels, suggesting that the usefulness of FDP or D-dimer for the diagnosis of DIC may be decreased in hematological malignancy [22]. Although the AUC of FDP was markedly high for both DIC and pre-DIC, the AUC of the platelet count and PT-INR was markedly high for DIC but not for pre-DIC. The AUC of sCLEC-2 was moderately high for both DIC and pre-DIC, suggesting that sCLEC-2 may be useful for the diagnosis of pre-DIC and early-stage DIC.

Regarding the formula of sCLEC-2, sCLEC-2/PLT or sCLEC-2xD-dimer increased the diagnostic for DIC or pre-DIC in comparison to sCLEC-2 alone, and sCLEC-2xD-dimer/PLT was the most useful for the diagnosis of both DIC and pre-DIC. As thrombocytopenia is frequently observed in DIC, the levels of plasma sCLEC-2 released from decreased platelets may not be sufficiently elevated in patients with DIC. Accordingly, the sCLEC-2/PLT ratio has been reported to be useful for the diagnosis of DIC [43]. An elevated sCLEC-2/PLT ratio was also reported in postoperative glioma patients with venous thromboembolism [44]. As D-dimer and FDP were the most useful biomarkers for DIC in this study [45], sCLEC-2xD-dimer, which reflects the activation of both platelets and coagulation, showed high diagnostic ability for DIC. In particular, the sCLEC-2xD-dimer/PLT ratio was more useful for the diagnosis of pre-DIC in comparison to FDP or D-dimer because the treatment of early phase DIC (e.g., pre-DIC or SIC) is recommended [23,26]. 

Regarding the outcome, the DIC score, FDP, sCLEC-2xD-dimer/PLT, and PT-INR were useful biomarkers for predicting a poor outcome. The complication of DIC is considered to be associated with poor outcomes [22]. The outcomes of DIC diagnosed using the JMHLW criteria were poorer in comparison to DIC diagnosed using JAAM [22], suggesting that the diagnostic criteria for the early phase of DIC cannot sufficiently predict a poor outcome. ROC analysis showed that the cutoff value of the JMHLW DIC score for a poor outcome was 3.3. These findings suggest that the treatment of DIC for the improvement of an outcome should start at a DIC score of four points before the diagnosis of DIC (DIC score of seven points). In addition, the outcome generally depends on underlying diseases; thus, survival should be examined in a large-scale study. 

Although the comparison of the accuracy of the JMHLW diagnostic criteria and the super formula using sCLEC-2 in the diagnosis of DIC requires a gold standard definition for DIC, the super formula using sCLEC-2 is simpler and easier to apply than JMHLW diagnostic criteria. In addition, the present study demonstrates that the concordance between the JMHLW diagnostic criteria and the super formula using sCLEC-2 is significantly high and that the super formula using sCLEC-2 can be used in place of the JMHWD diagnostic criteria in the diagnosis of DIC.

## 5. Conclusions

sCLEC-2xD-dimer/PLT, which can be diagnosed without a complicated scoring system, is a simple and useful diagnostic formula for the diagnosis of pre-DIC as well as overt-DIC. 

## Figures and Tables

**Figure 1 diagnostics-13-02299-f001:**
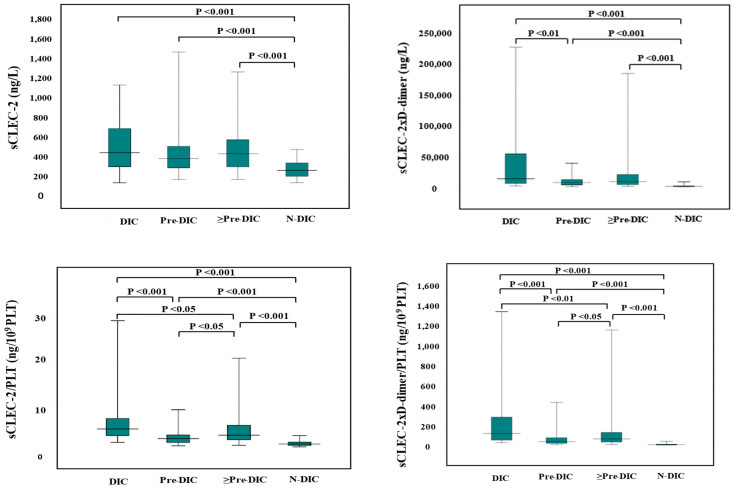
sCLEC-2, sCLEC-2xD-dimer, sCLEC-2/PLT, and sCLEC-2xD-dimer/PLT in DIC, pre-DIC, ≥Pre-DIC and N-DIC. DIC, disseminated intravascular coagulation; PLT, platelet count; sCLEC-2, soluble C-type lectin-like receptor 2; ≥Pre-DIC, DIC + pre-DIC; N-DIC, non-DIC.

**Figure 2 diagnostics-13-02299-f002:**
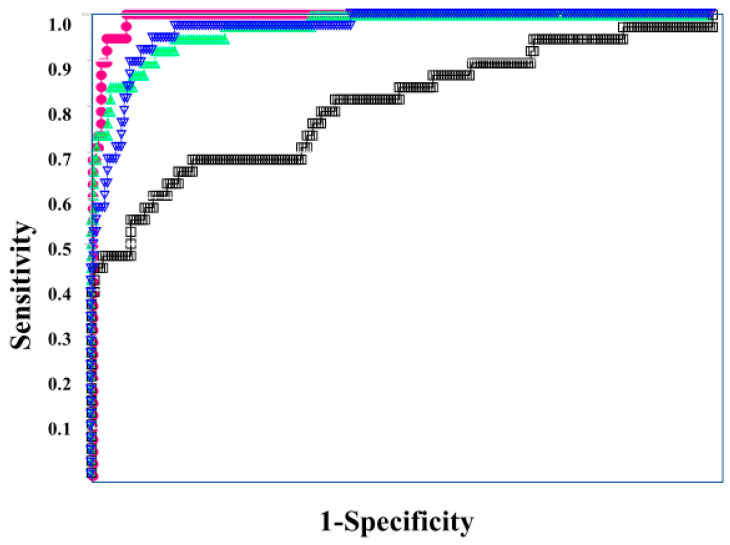
The ROC analysis of sCLEC-2, sCLEC-2xD-dimer, sCLEC-2/PLT, and sCLEC-2xD-dimer/PLT for DIC vs. N-DIC. ROC, receiver operating characteristic; DIC, disseminated intravascular coagulation; PLT, platelet count; sCLEC-2, soluble C-type lectin-like receptor 2; N-DIC, non-DIC. sCLEC-2 (**□**); sCLEC-2xD-dimer (**▽**); sCLEC-2/PLT (**△**); sCLEC-2xD-dimer/PLT (**●**).

**Figure 3 diagnostics-13-02299-f003:**
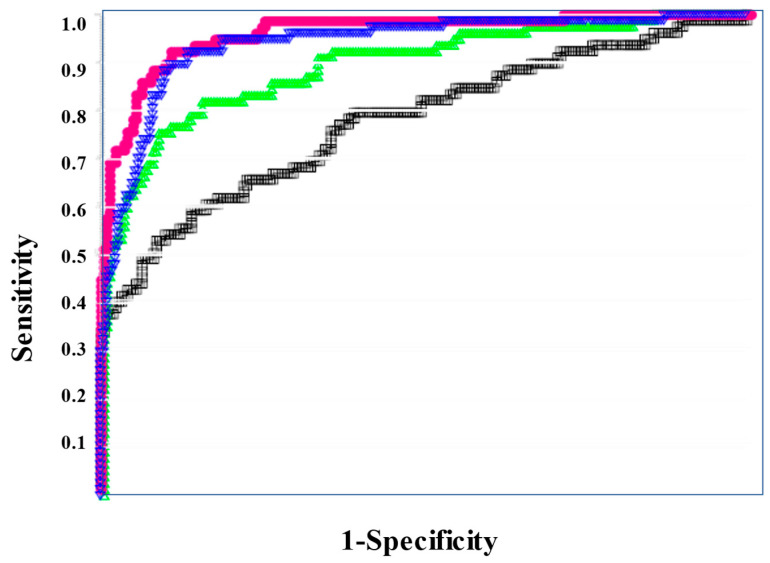
The ROC analysis of sCLEC-2, sCLEC-2xD-dimer, sCLEC-2/PLT, and sCLEC-2xD-dimer/PLT for ≥Pre-DIC vs. N-DIC. ROC, receiver operating characteristic; DIC, disseminated intravascular coagulation; PLT, platelet count; sCLEC-2, soluble C-type lectin-like receptor 2; ≥Pre-DIC, DIC + pre-DIC; N-DIC, non-DIC; sCLEC-2 (**□**); sCLEC-2xD-dimer (**▽**); sCLEC-2/PLT (**△**); sCLEC-2xD-dimer/PLT (**●**).

**Figure 4 diagnostics-13-02299-f004:**
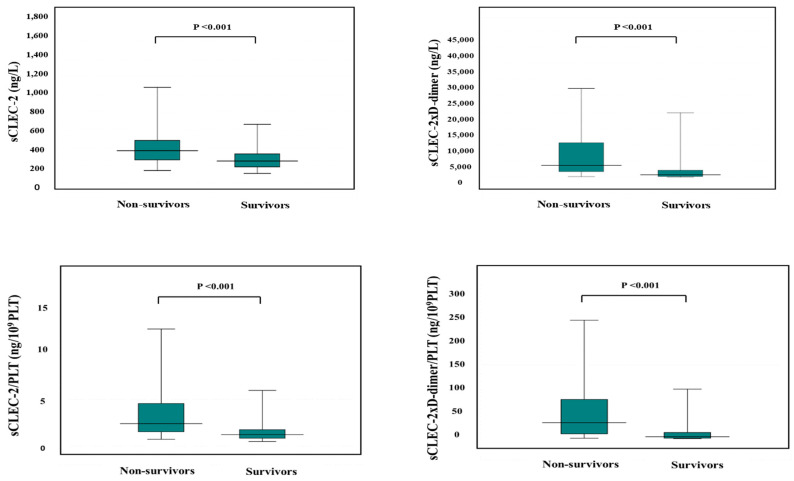
sCLEC-2, sCLEC-2xD-dimer, sCLEC-2/PLT, and sCLEC-2xD-dimer/PLT in non-survivors and survivors. PLT, platelet count; sCLEC-2, soluble C-type lectin-like receptor 2.

**Figure 5 diagnostics-13-02299-f005:**
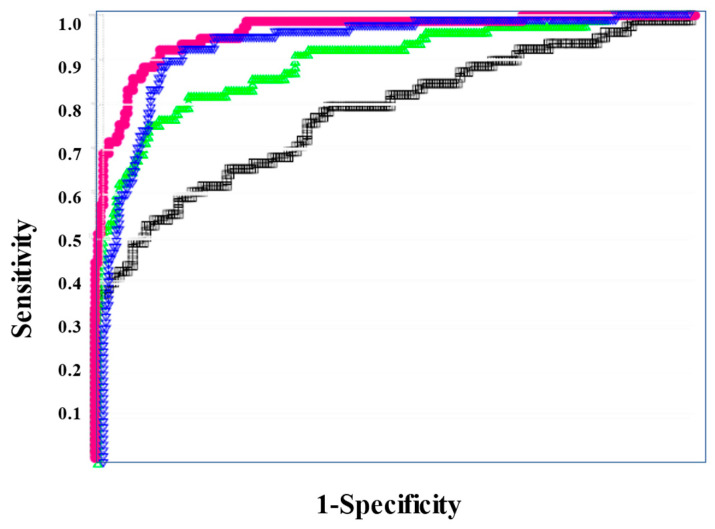
ROC analysis of sCLEC-2, sCLEC-2xD-dimer, sCLEC-2/PLT, and sCLEC-2xD-dimer/PLT for non-survivors vs. survivors. ROC, receiver operating characteristic; DIC, disseminated intravascular coagulation; PLT, platelet count; sCLEC-2 (**□**), soluble C-type lectin-like receptor 2. sCLEC-2xD-dimer (**▽**); sCLEC-2/PLT (**△**); sCLEC-2xD-dimer/PLT (**●**).

**Table 1 diagnostics-13-02299-t001:** The subject study protocol (2019-K9) was approved by the Human Ethics Review Committee of Mie Prefectural General Medical Center, and informed consent was obtained from each participant. This study was carried out in accordance with the principles of the Declaration of Helsinki.

	Solid Cancer	HM	AA	Trauma	Infection	CPA	UCS
*n*	27	11	37	41	215	25	74
Age	73.0	73.0	74.0	69.0	77.0	81.0	57.5
(years old)	(69.0–79.8)	(58.0–83.0)	(67.0–78.0)	(42.0–80.3)	(61.0–83.0)	(68.8–87.3)	(48.0–73.0)
Sex (F:M)	12:15	4:7	17:20	20:21	99:116	8:17	39:35
Death (Mortality)	8 (29.6%)	0 (0%)	3 (8.1%)	4 (9.8%)	25 (11.6%)	22 (88%)	0 (0%)
APTT (sec)	30.5	30.0	29.5	28.0	33.0 ***	57.0 ***	29.0
	(27.0–35.0)	(27.0–35.8)	(28.0–34.5)	(26.0–34.0)	(29.0–39.0)	(44.3–76.0)	(27.0–32.0)
PT-INR	1.07 ***	0.97	1.02 ***	0.96	1.13 ***	1.62 ***	0.96
	(1.02–1.21)	(0.94–1.26)	(0.97–1.18)	(0.92–1.05)	(1.03–1.24)	(1.21–1.97)	(0.92–1.00)
PLT (×10^9^/L)	226	191	160 ***	227	191 ***	114 ***	23.4
	(137–309)	(132–258)	(119–200)	(166–275)	(125–253)	(85–178)	(188–275)
DIC score	2.0 ***	1.0 ***	3.0 ***	1.0 ***	2.0 ***	7.0 ***	0
	(1.0–4.8)	(1.0–3.8)	(1.0–4.0)	(1.0–4.0)	(1.0–4.0)	(5.0–8.0)	(0–0)
FDP	4.4 ***	4.6 **	3.7 ***	8.5 ***	5.6 ***	67.6 ***	0.7
(μg/mL)	(2.8–28.3)	(1.2–17.9)	(0.6–7.5)	(3.3–37.8)	(2.4–14.5)	(22.2–452.8)	(0.3–1.0)
D-dimer (μg/mL)	3.8 ***	4.1 **	6.8 ***	7.6 ***	4.4 ***	15.7 ***	0.6
	(1.9–16.6)	(0.7–11.9)	(2.6–11.6)	(2.4–18.0)	(1.6–9.7)	(7.6–46.4)	(0.4–1.5)
SF	6.7 ***	5.8	8.1 ***	6.8 ***	9.6 ***	12.7 ***	2.1
(μg/mL)	(2.2–13.4)	(3.0–21.1)	(1.6–12.7)	(1.1–9.1)	(2.0–15.0)	(2.1–25.5)	(0.5–4.7)
sCLEC2	260 **	278 *	247 ***	245 ***	258 **	441 ***	193
(ng/L)	(172–321)	(167–364)	(174–323)	(178–328)	(195–335)	(310–748)	(143–242)

Data are shown as median (25–75 percentile). HM, hematological malignancy; AA, aortic aneurysm; CPA, cardiopulmonary arrest; UCS, unidentified clinical syndrome; APTT, activated partial thromboplastin time; PT-INR, prothrombin time-international normalized ratio; PLT, platelet count; DIC, disseminated intravascular coagulation; FDP, fibrinogen and fibrin degradation products; SF, soluble fibrin; sCLEC-2, soluble C-type lectin-like receptor 2; *, *p* < 0.05; **, *p* < 0.01; and ***, *p* < 0.001 in comparison to UCS.

**Table 2 diagnostics-13-02299-t002:** Hemostatic markers in patients with DIC, pre-DIC, or non-DIC.

	DIC (*n* = 38)	Pre-DIC (*n* = 39)	DIC + Pre-DIC (*n*= 77)	Non-DIC (*n* = 222)
FDP (μg/mL)	70.5 (35.5–435.9) ***	27.8 (16.7–49.9) ***	43.2 (22.9–85.1) ***	3.9 (1.8–7.5)
D-dimer (μg/mL)	27.9 (16.6–74.9) ***	18.5 (7.9–29.3) ***	22.1 (11.6–41.1) ***	3.0 (1.2–6.0)
PT-INR	1.69 (1.38–1.98) ***	1.21 (1.02–1.40) ***	1.39 (1.15–1.74) ***	1.05 (0.97–1.13)
PLT (×10^9^/L)	93 (45–124) ***	156 (111–241) ***	119 (85–200) ***	214 (156–270)
SF (μg/mL)	49.1 (28.3–98.6) ***	26.5 (15.8–57.2) ***	35.3 (17.5–69.1) ***	11.1 (6.9–20.2)
APTT (sec)	57.5 (47.5–78.3) ***	35.0 (29.5–45.8) ***	46.0 (33.0–65.0) ***	30.0 (27.0–34.0)
sCLEC2 (ng/L)	423 (275–679) ***	361 (264–492) ***	413 (274–560) ***	237 (174–317) ***

Data are shown as median (25–75 percentile). DIC, disseminated intravascular coagulation; FDP, fibrinogen and fibrin degradation products; PT-INR, prothrombin time-international normalized ratio; PLT, platelet count; SF, soluble fibrin; APTT, activated partial thromboplastin time; sCLEC-2, soluble C-type lectin-like receptor 2; and ***, *p* < 0.001 in comparison to non-DIC.

**Table 3 diagnostics-13-02299-t003:** The ROC analysis of hemostatic biomarkers for the diagnosis of DIC, pre-DIC, or DIC + pre-DIC vs. non-DIC.

		DIC	Pre-DIC	DIC + Pre-DIC
FDP	Cutoff value (μg/mL)	21.8	12.5	16.0
	Sensitivity (specificity)	91.9%	84.6%	86.5%
	AUC	0.972	0.894	0.933
	Odds ratio	132.2	30.4	42.9
D-dimer	Cutoff value (μg/mL)	10.4	7.0	8.3
	Sensitivity (specificity)	90.1%	79.5%	84.4%
	AUC	0.960	0.883	0.921
	Odds ratio	77.3	15.2	30.0
PT-INR	Cutoff value	1.20	1.08	1.13
	Sensitivity (specificity)	88.2%	64.7%	76.0%
	AUC	0.943	0.701	0.821
	Odds ratio	48.0	3.3	9.8
PLT	Cutoff value (×10^9^/L)	137	190	170
	Sensitivity (specificity)	81.6%	59.0%	68.8%
	AUC	0.890	0.652	0.758
	Odds ratio	20.2	2.1	4.9
SF	Cutoff value (μg/mL)	24.1	16.2	19.1
	Sensitivity (specificity)	81.6%	66.3%	74.0%
	AUC	0.905	0.749	0.819
	Odds ratio	27.9	3.9	8.3
APTT	Cutoff value	35.1	31.4	33.0
	Sensitivity (specificity)	83.4%	64.5%	73.8%
	AUC	0.911	0.670	0.787
	Odds ratio	25.4	3.1	7.9
sCLEC-2	Cutoff value (ng/L)	287	284	285
	Sensitivity (specificity)	68.4%	67.6%	67.6%
	AUC	0.801	0.748	0.774
	Odds ratio	4.7	4.2	4.4

ROC, receiver operating characteristic; DIC, disseminated intravascular coagulation; FDP, fibrinogen and fibrin degradation products; PT-INR, prothrombin time-international normalized ratio; PLT, platelet count; SF, soluble fibrin; APTT, activated partial thromboplastin time; sCLEC-2, soluble C-type lectin-like receptor 2; AUC, area under the curve.

**Table 4 diagnostics-13-02299-t004:** sCLEC-2, sCLEC-2/PLT, sCLEC-2xD-dimer, and sCLEC-2xD-dimer/PLT.

	DIC	Pre-DIC	DIC + Pre-DIC	Non-DIC
sCLEC-2(ng/L)	423 (275–679) ***	361 (264–492) ***	413 (274–560) ***	237 (174–317)
sCLEC-2/PLT	4.78 (3.05–7.05) ***	2.37 (1.47–3.21) ***	3.16 (2.12–5.60) ***	1.12 (0.82–1.55)
sCLEC-2xD-dimer	13,198 (5255–54,369) ***	6923 (2906–11,842) ***	8533 (3467–20,348) ***	657 (254–1542)
sCLEC-2xD-dimer/PLT	128 (53.0–509) ***	35.6 (19.5–761) ***	65.4 (33.2–141) ***	3.45 (1.21–7.35)

DIC, disseminated intravascular coagulation; PLT, platelet count; sCLEC-2, soluble C-type lectin-like receptor 2; ***, *p* < 0.001 in comparison to non-DIC.

**Table 5 diagnostics-13-02299-t005:** The ROC analysis of sCLEC-2, sCLEC2/PLT, sCLEC2xD-dimer, and sCLEC-2xD-dimer/PLT for the diagnosis of DIC, pre-DIC, or DIC + pre-DIC vs. non-DIC.

		DIC	Pre-DIC	DIC + Pre-DIC
sCLEC-2	Cutoff value (ng/L)	287	284	285
	Sensitivity (specificity)	68.4%	67.6%	67.6%
	AUC	0.801	0.748	0.774
	Odds ratio	4.7	4.2	4.4
sCLEC-2/PLT	Cutoff value	2.07	1.54	1.74
	Sensitivity (specificity)	89.6%	73.9%	81.8%
	AUC	0.970	0.822	0.895
	Odds ratio	100.9	6.4	20.5
sCLEC-2xD-dimer	Cutoff value	2993	2252	2429
	Sensitivity (specificity)	91.9%	86.5%	89.2%
	AUC	0.966	0.911	0.938
	Odds ratio	132.2	43.5	71.2
sCLEC-2xD-dimer/PLT	Cutoff value	25.6	13.1	17.0
	Sensitivity (specificity)	94.7%	85.6%	89.6%
	AUC	0.993	0.929	0.961
	Odds ratio	315.0	32.7	74.6

ROC, receiver operating characteristic; DIC, disseminated intravascular coagulation; PLT, platelet count; sCLEC-2, soluble C-type lectin-like receptor 2; AUC, area under the curve.

**Table 6 diagnostics-13-02299-t006:** Hemostatic markers in survivors and non-survivors.

	Survivors	Non-Survivors
APTT (sec)	31.0 (27.0–34.0) ***	46.0 (33.0–65.0) ***
PT-INR	1.06 (0.97–1.16) ***	1.40 (1.13–1.87) ***
PLT (×10^9^/L)	200 (138–267) ***	134 (90–207) ***
FDP (μg/mL)	3.5 (16.0–8.6) ***	49.9 (16.1–179.4) ***
DIC score	2.0 (1.0–3.0) ***	6.0 (4.0–8.0) ***
D-dimer LG (μg/mL)	3.8 (1.4–8.3) ***	16.8 (7.3–33.2) ***
SF (μg/mL)	12.7 (7.5–23.8) ***	30.2 (14.5–56.2) ***
sCLEC2 (ng/L)	247 (180–328) ***	365 (261–480) ***
sCLEC-2/PLT	1.25 (0.85–1.77) ***	2.53 (1.55–5.04) ***
sCLEC-2xD-dimer	771 (280–2254) ***	4962 (1934–13,624) ***
sCLEC-2xD-dimer/PLT	4.29 (1.24–13.2) ***	46.3 (15.6–128) ***

Data are shown as median (25–75 percentile). DIC, disseminated intravascular coagulation; FDP, fibrinogen and fibrin degradation products; PT-INR, prothrombin time-international normalized ratio; PLT, platelet count; SF, soluble fibrin; APTT, activated partial thromboplastin time; sCLEC-2, soluble C-type lectin-like receptor 2; ***, *p* < 0.001 between survivors and non-survivors.

**Table 7 diagnostics-13-02299-t007:** The ROC analysis of hemostatic biomarkers for non-survivors vs. survivors.

FDP	Cutoff value (μg/mL)	14.7
	Sensitivity (specificity)	76.8%
	AUC	0.840
	Odds ratio	11.0
D-dimer	Cutoff value (μg/mL)	7.9
	Sensitivity (specificity)	74.5%
	AUC	0.786
	Odds ratio	8.0
PT-INR	Cutoff value	1.14
	Sensitivity (specificity)	70.6%
	AUC	0.817
	Odds ratio	5.5
PLT	Cutoff value (×10^9^/L)	171
	Sensitivity (specificity)	63.2%
	AUC	0.701
	Odds ratio	2.9
DIC score	Cutoff value (×10^10^/L)	3.3
	Sensitivity (specificity)	79.3%
	AUC	0.847
	Odds ratio	15.0
SF	Cutoff value (μg/mL)	18.3
	Sensitivity (specificity)	66.1%
	AUC	0.709
	Odds ratio	3.8
APTT	Cutoff value (μg/mL)	33.5
	Sensitivity (specificity)	72.7%
	AUC	0.768
	Odds ratio	6.2
sCLEC2	Cutoff value (ng/L)	286
	Sensitivity (specificity)	64.3%
	AUC	0.695
	Odds ratio	3.2
sCLEC-2/PLT	Cutoff value	1.67
	Sensitivity (specificity)	71.4%
	AUC	0.781
	Odds ratio	6.3
sCLEC-2xD-dimer	Cutoff value	2041
	Sensitivity (specificity)	72.8%
	AUC	0.798
	Odds ratio	7.3
sCLEC-2xD-dimer/PLT	Cutoff value	14.4
	Sensitivity (specificity)	76.8%
	AUC	0.816
	Odds ratio	11.0

ROC, receiver operating characteristic; DIC, disseminated intravascular coagulation; FDP, fibrinogen and fibrin degradation products; PT-INR, prothrombin time-international normalized ratio; PLT, platelet count; SF, soluble fibrin; APTT, activated partial thromboplastin time; sCLEC-2, soluble C-type lectin-like receptor 2; AUC, area under the curve.

## Data Availability

The data presented in this study are available on request from the corresponding author. The data are not publicly available due to privacy restrictions.

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
