# Peer review of "Super Formula for Diagnosing Disseminated Intravascular Coagulation Using Soluble C-Type Lectin-like Receptor 2"

_diagnostics, 2023, doi:10.3390/diagnostics13132299_

Round 1
Reviewer 1 Report
very meticulous and important work.
The methods were sound and findings valid
Author Response
Comment
very meticulous and important work.
The methods were sound and findings valid.
Response. Thank you for your warm comment. This manuscript has been carefully revised.
Reviewer 2 Report
In this manuscript, the authors retrospectively evaluated soluble C-type lectin like receptor 2 (sCLEC-2) and other hemostatic biomarkers in patients with DIC, pre-DIC and patients without DIC /pre-DIC. The authors found the sCLEC-2xD-dimer/PLT as a simple and useful formula for the diagnosis of pre-DIC and DIC. The data are extensive, the findings are significant, and conclusion is supported by the results.
Major:
The authors found the formula sCLEC-2xD-dimer/PLT is simple than other frequently used diagnostic criteria for DIC. Since JMHLW is the most frequent used criteria for DIC in Japan, the authors should also compare or discuss the accuracy between JMHLW and the proposed novel formula in DIC diagnosis.
The authors mentioned that they evaluated the criteria of 39 patients with DIC, 38 patients with pre-DIC and 222 patients without DIC in the abstract. However, in table 2, the number of patients with DIC, pre-DIC and non-DIC in this study were 38, 39 and 222, respectively. The authors should further confirm the number of patients enrolled in this study.
Minor:
The font size of the letters in line 161, page 6 seem to be smaller than that of the other part.
Line 61, page 2, “sepsis-induced”.
An extra blank in line 111, page 4.
Minor editing of English language required
Author Response
Major:
Comment 1
The authors found the formula sCLEC-2xD-dimer/PLT is simple than other frequently used diagnostic criteria for DIC. Since JMHLW is the most frequent used criteria for DIC in Japan, the authors should also compare or discuss the accuracy between JMHLW and the proposed novel formula in DIC diagnosis.
Response 1. Although the comparison of accuracy of the JMHLW diagnostic criteria and the super formula using sCLEC-2 in the diagnosis of DIC requires a gold-standard for DIC, the super formula using sCLEC-2 is simpler and easier to apply than the JMHLW diagnostic criteria. In addition, the present study demonstrates that the concordance between the JMHLW diagnostic criteria and the super formula using sCLEC-2 is significantly high and that the super formula using sCLEC-2 can be used in place of the JMHLW diagnostic criteria in the diagnosis of DIC.
Comment 2
The authors mentioned that they evaluated the criteria of 39 patients with DIC, 38 patients with pre-DIC and 222 patients without DIC in the abstract. However, in table 2, the number of patients with DIC, pre-DIC and non-DIC in this study were 38, 39 and 222, respectively. The authors should further confirm the number of patients enrolled in this study.
Response 2. The abstract has been revised as follows “---among 38 patients with DIC, 39 patients with pre-DIC---".
Minor:
Comment 3
The font size of the letters in line 161, page 6 seem to be smaller than that of the other part.
Response 3. The font size has been corrected.
Comment 4
Line 61, page 2, “sepsis-induced”.
Response 4. These words have been revised.
Comment 5
An extra blank in line 111, page 4.
Response 5. This line has been revised.
Comment 6
Minor editing of English language required
Response 6. The manuscript has been revised by a professional editor who is a native speaker of English.
Reviewer 3 Report
A relevant dissertation topic. Focused and concise.
Author Response
Comment
A relevant dissertation topic. Focused and concise.
Response. Thank you for your warm comment. This manuscript has been carefully revised.
Reviewer 4 Report
The presented work has been done clearly and provides appropriate results. It is relevant for the field, scientifically sound, and presented in a well-structured manner. The performed statistical analysis is convincing. In the article, the platelet count is presented as n x 1010/l instead of the standard n x 109/l, which can lead to an error in its estimation. So, for a long time, I wondered why all the study participants even without DIC had such severe thrombocytopenia. In order to avoid causing this confusion among inattentive readers, the number of platelets may be corrected for the generally accepted designation. The text contains several typos (4 in one sentence, line 132). In general, I have no significant remarks. As a result, the manuscript may be recommended for publication.Author Response
Comment
The presented work has been done clearly and provides appropriate results. It is relevant for the field, scientifically sound, and presented in a well-structured manner. The performed statistical analysis is convincing. In the article, the platelet count is presented as n x 1010/l instead of the standard n x 109/l, which can lead to an error in its estimation. So, for a long time, I wondered why all the study participants even without DIC had such severe thrombocytopenia. In order to avoid causing this confusion among inattentive readers, the number of platelets may be corrected for the generally accepted designation. The text contains several typos (4 in one sentence, line 132). In general, I have no significant remarks. As a result, the manuscript may be recommended for publication.
Response. The platelet count was changed to n x 109/l from n x 1010/l. This manuscript has been carefully revised. All of the tables, as well as figures 1 and 4 have been revised accordingly.